# T-Cell Lymphoma Clonality by Copy Number Variation Analysis of T-Cell Receptor Genes

**DOI:** 10.3390/cancers13020340

**Published:** 2021-01-19

**Authors:** Ming Liang Oon, Jing Quan Lim, Bernett Lee, Sai Mun Leong, Gwyneth Shook-Ting Soon, Zi Wei Wong, Evelyn Huizi Lim, Zhenhua Li, Allen Eng Juh Yeoh, Shangying Chen, Kenneth Hon Kim Ban, Tae-Hoon Chung, Soo-Yong Tan, Shih-Sung Chuang, Seiichi Kato, Shigeo Nakamura, Emiko Takahashi, Yong-Howe Ho, Joseph D. Khoury, Rex K. H. Au-Yeung, Chee-Leong Cheng, Soon-Thye Lim, Wee-Joo Chng, Claudio Tripodo, Olaf Rotzschke, Choon Kiat Ong, Siok-Bian Ng

**Affiliations:** 1Department of Pathology, National University Hospital, National University Health System, Singapore 119074, Singapore; ming_liang_oon@nuhs.edu.sg (M.L.O.); gwyneth_st_soon@nuhs.edu.sg (G.S.-T.S.); zi_wei_wong@nuhs.edu.sg (Z.W.W.); pattsy@nus.edu.sg (S.-Y.T.); 2Lymphoma Genomic Translational Research Laboratory, Division of Cellular and Molecular Research, National Cancer Centre Singapore, Singapore 169610, Singapore; lim.jing.quan@nccs.com.sg; 3Cancer and Stem Cell Biology, Duke-NUS Medical School, Singapore 169857, Singapore; 4Singapore Immunology Network (SIgN), A*STAR (Agency for Science, Technology and Research), Singapore 138632, Singapore; bernett_lee@immunol.a-star.edu.sg (B.L.); olaf_rotzschke@immunol.a-star.edu.sg (O.R.); 5Department of Pathology, Yong Loo Lin School of Medicine, National University of Singapore, Singapore 119074, Singapore; patlsm@nus.edu.sg; 6Viva-NUS Centre for Translational Research in Acute Leukaemia, Department of Paediatrics, Yong Loo Lin School of Medicine, National University of Singapore, Singapore 117597, Singapore; paelhe@nus.edu.sg (E.H.L.); paeliz@nus.edu.sg (Z.L.); allen_yeoh@nuhs.edu.sg (A.E.J.Y.); 7VIVA—University Children’s Cancer Centre, Khoo Teck Puat–National University Children’s Medical Institute, National University Hospital, National University Health System, Singapore 119074, Singapore; 8Department of Biochemistry, Yong Loo Lin School of Medicine, National University of Singapore, Singapore 117596, Singapore; bchchen@nus.edu.sg (S.C.); bchbhkk@nus.edu.sg (K.H.K.B.); 9Cancer Science Institute of Singapore, National University of Singapore, Singapore 117599, Singapore; csicth@nus.edu.sg (T.-H.C.); mdccwj@nus.edu.sg (W.-J.C.); 10Department of Pathology, Chi-Mei Medical Center, Tainan 71004, Taiwan; cmh5301@mail.chimei.org.tw; 11Department of Pathology and Laboratory Medicine, Nagoya University Hospital, Nagoya 466-8560, Japan; skato@aichi-cc.jp (S.K.); snakamur@med.nagoya-u.ac.jp (S.N.); 12Department of Pathology and Molecular Diagnostics, Aichi Cancer Center Hospital, Nagoya 464-0021, Japan; 13Department of Pathology, Aichi Medical University Hospital, Nagakute 480-1195, Japan; emikot@aichi-med-u.ac.jp; 14Department of Pathology, Tan Tock Seng Hospital, Singapore 308433, Singapore; yong_howe_ho@ttsh.com.sg; 15Department of Hematopathology, The University of Texas MD Anderson Cancer Center, Houston, TX 77030, USA; jkhoury@mdanderson.org; 16Department of Pathology, Queen Mary Hospital, The University of Hong Kong, Hong Kong, China; rex.auyeung@hku.hk; 17Department of Pathology, Singapore General Hospital, Singapore 169608, Singapore; cheng.chee.leong@singhealth.com.sg; 18Lymphoma Genomic Translational Research Laboratory, Division of Medical Oncology, National Cancer Centre Singapore, Singapore 169610, Singapore; lim.soon.thye@singhealth.com.sg; 19Department of Hematology-Oncology, National University Cancer Institute Singapore, National University Hospital, National University Health System, Singapore 119074, Singapore; 20Tumor Immunology Unit, University of Palermo School of Medicine, 90134 Palermo, Italy; claudio.tripodo@unipa.it; 21Genome Institute of Singapore, A*STAR (Agency for Science, Technology and Research), Singapore 138632, Singapore

**Keywords:** whole genome sequencing, T-cell receptor, clonality, copy number variation analysis, T-cell lymphoma

## Abstract

**Simple Summary:**

T-cells defend the human body from pathogenic invasion via specific recognition by T-cell receptors (TCRs). The TCR genes undergo recombination (rearrangement) in a myriad of possible ways to generate different TCRs that can recognize a wide diversity of foreign antigens. However, in patients with T-cell lymphoma (TCL), a particular T-cell becomes malignant and proliferates, resulting in a population of genetically identical cells with same TCR rearrangement pattern. To help diagnose patients with TCL, a polymerase chain reaction (PCR)-based assay is currently used to determine if neoplastic cells in patient samples are of T-cell origin and bear identical (monoclonal) TCR rearrangement pattern. Herein, we report the application of a novel segmentation and copy number computation algorithm to accurately identify different TCR rearrangement patterns using data from the whole genome sequencing of patient materials. Our approach may improve the diagnostic accuracy of TCLs and can be similarly applied to the diagnosis of B-cell lymphomas.

**Abstract:**

T-cell lymphomas arise from a single neoplastic clone and exhibit identical patterns of deletions in T-cell receptor (TCR) genes. Whole genome sequencing (WGS) data represent a treasure trove of information for the development of novel clinical applications. However, the use of WGS to identify clonal T-cell proliferations has not been systematically studied. In this study, based on WGS data, we identified monoclonal rearrangements (MRs) of T-cell receptors (TCR) genes using a novel segmentation algorithm and copy number computation. We evaluated the feasibility of this technique as a marker of T-cell clonality using T-cell lymphomas (TCL, *n* = 44) and extranodal NK/T-cell lymphomas (ENKTLs, *n* = 20), and identified 98% of TCLs with one or more TCR gene MRs, against 91% detected using PCR. TCR MRs were absent in all ENKTLs and NK cell lines. Sensitivity-wise, this platform is sufficiently competent, with MRs detected in the majority of samples with tumor content under 25% and it can also distinguish monoallelic from biallelic MRs. Understanding the copy number landscape of TCR using WGS data may engender new diagnostic applications in hematolymphoid pathology, which can be readily adapted to the analysis of B-cell receptor loci for B-cell clonality determination.

## 1. Introduction

The T-cell receptor (TCR) is a heterodimeric glycoprotein existing as two molecules, TCRαβ or TCRγδ. In TCRαβ, a TCRα chain is linked to a TCRβ chain via a disulfide bond, whereas in TCRγδ, a TCRγ chain is linked to a TCRδ chain [1]. The generation of diversity in T-cell receptor (TCR) genes is a critical component of adaptive immunity, achieved through somatic recombination of multiple variable (V), diversity (D), and joining (J) regions, mediated by RAG recombinase. Rearrangement occurs in a sequential fashion, beginning with TCR delta (TCRD), followed by gamma (TCRG), beta (TCRB), and alpha (TCRA). Like the immunoglobulin light chain, TCRA and TCRG loci contain only V and J gene segments, while the TCRB and TCRD loci contain all three gene segments (V, D, and J), which is analogous to the immunoglobulin heavy chain [1]. Isotype switching and somatic hypermutation, processes that generate diversity in antigen-activated B-cells, do not occur in T-cells. Instead, the diversity in T-cell repertoire is established before antigenic stimulation through somatic recombination [1]. Sequences between the rearranged sequences are deleted [2,3,4]. T-cell lymphomas (TCLs) arise from a proliferation of a neoplastic T-cell clone with identical TCR gene rearrangements and the presence of monoclonal TCR rearrangements in TCLs is conventionally established through polymerase chain reaction (PCR) [3,5]. However, PCR-based assays face limitations with restricted coverage of the TCR gene owing to amplicon size limitations, impeding interrogation of the larger TCRA locus [2,5]. We have previously noted unique patterns of copy number (CN) losses in the TCRA gene on the OncoScan platform, which provided us with the impetus to perform a systematic copy number variation (CNV) analysis on a larger cohort using whole genome sequencing (WGS) [6]. We hypothesize that TCLs should exhibit identical patterns of deletions within the TCR gene loci. Using a novel segmentation algorithm and CN computation, we were able to identify the monoclonal rearrangements (MRs) of the TCR genes and we further evaluated the feasibility of this novel technique as a marker of T-cell clonality.

## 2. Results

### 2.1. CNV Analysis of TCR and Ig Loci in TCLs and ENKTL

Using our CNV-based approach, 98% (43/44) TCLs showed monoclonal losses of one or more TCR loci, of which MRs of TCRG were the most common (93%, 41/44), followed by TCRB (89%, 39/44), TCRA (82%, 36/44), and TCRD (18%, 8/44) (Table 1). TCR deletions were absent in all ENKTLs and cell lines of NK lineage and present in three of four ENKTLs of T-cell lineage (Appendix A). In comparison, PCR detected monoclonality in 91% (28/31) of TCLs with MRs of TCRG and TCRB in 84% (26/31) and 77% (24/31) of cases, respectively (Table 1 and Table 2). Of 31 cases with both WGS and PCR data for comparison, WGS detected MRs of TCRG (*n* = 4) and TCRB (*n* = 6) not identified by PCR (Table 2, Figure 1). This is not unexpected since WGS is able to provide entire coverage of the TCR loci while PCR sensitivity is dependent on primer designs, which may not be able to detect all possible monoclonal rearrangements. Conversely, PCR also detected MRs of TCRG (*n* = 1) and TCRB (*n* = 2) which were not recognized by our WGS method. The possible reasons for the false negative result using our WGS-CNV approach may be attributed to (i) low tumor content and (ii) very small deletions (~1 kbp) which may not be detectable by current segmentation algorithms. 

The allelic status of the monoclonal rearrangement (monoallelic or biallelic) could be determined using WGS in 41 cases for TCRG and 34 cases for TCRB. Out of these cases, biallelic rearrangements were identified in 51% (21/41) of TCRG rearrangements and 27% (9/34) of TCRB rearrangements. In comparison, determination of allelic status using PCR was possible in 26 cases for TCRG and 20 cases for TCRB, of which biallelic rearrangements were found in 58% (15/26) of TCRG rearrangements and 50% (10/20) of TCRB rearrangements (Table 1, Figure 2a). For monoallelic rearrangements, WGS had a detection rate of 49% (20/41) for TCRG and 76% (25/33) for TCRB, compared with 42% (11/26) and 50% (10/20), respectively, using PCR.

Amongst the samples of TCLs, eight cases (18%) showed monoclonal deletions of at least one of the BCR loci. Of the TCL cases, six cases were AITLs (30% of AITL), while two cases were PTCLs (17% of PTCL). Seven cases showed MRs of IgH, while one case showed MRs of IgL. In the ENKTL samples, four cases, including one cell line, showed monoclonal losses of the IgH locus (Appendix A). These findings are not unexpected, as monoclonal rearrangements of the BCR genes have been previously reported in cases of PTCL and ENKTL [2,7,8] and monoclonal BCR gene rearrangement can be found in as many as 20–30% of AITL cases [9].

The different patterns of losses of the TCR are presented in Figure 2b. A case is illustrated, herein, to elucidate our methodology (Figure 3a–c). Case 12, an angioimmunoblastic T-cell lymphoma with tumor content of 23%, showed MRs of TCRG, TCRB, and TCRA (Appendix A). As seen in the figure, segmented coverage plots of TCRG demonstrated sharply demarcated broad segment deletions identified as unique segments of the TCRG locus by the segmentation algorithm. The CN prediction correctly localized the CN data to the unique segments with a predicted CN of 0, corresponding to a biallelic loss (Figure 3a, left panel, green segment). The plot for TCRA (Figure 3b, left panel) shows multiple colored segments, representing a slopy (polyclonal) coverage depth across TCRA, characteristic of polyclonal loss (Figure 3b, left panel, indicated by blue brackets) resulting from contamination by background reactive T cells with a diverse pattern of TCR loss. This polyclonal pattern was followed by a flat segment of loss (Figure 3b, left panel, blue segment, indicated by green brackets) with predicted CN of 0 indicating monoclonal biallelic rearrangement of the TCRA gene occurring in a polyclonal background. The TCRB gene (Figure 3c, left panel) shows a copy number drop compared to the diploid state in the segment marked turquoise with monoallelic loss indicated by a copy number of 1. Another case (Figure 3d,e) of an angioimmunoblastic T-cell lymphoma (case 41) showed a very focal copy number loss (Figure 3d, circled area) of TCRA, which represents actual monoclonal loss of the TCRD gene (Figure 3e, green segment). Since TCRD resides within TCRA, copy number loss in the TCRD gene may be also be detectable in the TCRA locus.

We compared the patterns of losses of the TCR loci of different lymphoma subtypes (Appendix A). In the AITL, ALCL, and PTCL cases, a combination of TCRG, TCRB, and TCRA losses is the most common and present in 16/20 (80%), 4/6 (67%), and 8/12 (67%) of cases, respectively. This is not unexpected as the majority of T cell lymphomas are derived from αβ T-cells which have undergone all four TCR rearrangements. However, in most αβ T-cells, the TRD locus is deleted on both alleles, as during the organization of the locus, a functional TRA automatically leads to a deletion of the respective TRD, even if it was rearranged before [10], accounting for the lack of TCRD losses.

Additionally, we observed some cases which did not demonstrate the expected sequential pattern of TCR rearrangement (Figure 2c). Six out of 44 TCL cases showed rearrangement of TCRB in the absence of TCRA. Conversely, there were three cases with MRs of TCRA in the absence of TCRB rearrangements.

### 2.2. Tumor Content and In Silico Sensitivity Analysis

Tumor content of the samples ranged from 13% to 89% (Appendix A). Importantly, the majority of TCLs cases (94%, 17/18) with tumor content <25% showed rearrangements in at least one TCR gene (Appendix A), including a case with only 13% tumor content. We determined the correlation between the tumor content and sequencing depth with the number of TCR genes with monoclonal losses. There was no significant correlation between the number of monoclonal rearrangements that could be detected by WGS at low tumor concentration (Kruskal–Wallis test, *p* = 0.15) and sequencing depth (Kruskal–Wallis test, *p* = 0.32). To test the sensitivity of our methodology, WGS data were generated for a TCL cell line and a non-T (NK) cell line. The two sets of data were then merged at different proportions to obtain in silico dilutions of the TCL-cell lines at various concentrations (10% to 90%; at intervals of 10%). The MRs of TCRG and TCRD remained detectable even at 10% tumor content (Figure 4). However, the accuracy of the copy number prediction and the ability to differentiate monoallelic (CN = 0) from biallelic loss (CN = 0) is affected by the sequencing depth at low tumor concentration. With less sequencing depth, it becomes harder to distinguish between CN = 1 from CN = 0 and CN = 1 from CN = 2 when the sample has low tumor content. As illustrated in Figure 4, the biallelic loss (CN = 0, green segment) of TCRG appears as a monoallelic (CN = 1) loss when the concentration was reduced from 30% to 20%, even though the monoclonal loss remains detectable at 10% concentration.

### 2.3. TCRB Rearrangements

V-J rearrangements of the TCRB gene were detected in 34 out of 44 cases (77%), while D-J rearrangements were detected in 10 cases (23%). Short segments of D-J deletions could only be identified with careful analysis using a magnified view of the coverage plots, and could not be detected based on the whole TCRB gene segmentation plots. Targeted sequencing of TCRB was performed in selected TCL cases (*n* = 12) to validate the WGS findings, of which nine cases were positive for V-J and/or D-J TCRB rearrangements and three cases were negative by WGS (Appendix A). All three negative cases showed MRs of TCRB on targeted sequencing. It is noteworthy that tiny deletions arising from incomplete D-J TCRB rearrangement remained detectable using our technique. This is illustrated in Figure 3f,g, a case of an angioimmunoblastic T-cell lymphoma with an isolated incomplete rearrangement of the TCRB gene confirmed on sequencing. A coverage plot of the entire TCRB gene did not disclose any obvious monoclonal deletions. A magnified coverage plot of the DJ region of the TCRB gene (Figure 3g, corresponding to the region demarcated by the blue circle in Figure 3f) shows a broad segment deletion spanning gene segments TCRBD1 to TCRBJ2-7, corresponding to the incomplete TCRB rearrangement identified on sequencing. Such small deletions can be easily missed and should always be excluded when analyzing rearrangements of TCRB and TCRD genes where the process starts with a D to J rearrangement followed by a V to D-J rearrangement [3]. This is not so for TCRA and TCRG where direct V to J rearrangements occur and analysis for deletions in this context may be less problematic.

### 2.4. CNV Analysis of DLBCL Cell Lines

Finally, as a proof-of-concept that the proposed CNV analysis can also be applicable as a marker of clonality for BCLs, the same CNV analysis was performed on publicly available WGS data of 10 B-cell lymphoma (BCLs) cell lines and we were able to detect MRs of two or more BCR genes in all of them (Appendix A). Nine of these cases showed MRs of IgH, IgK, and IgL, while one case showed MRs of IgK and IgL. The patterns of the MRs of the Ig loci are illustrated in Figure 5. In addition, one case showed MRs of the TCRG gene.

## 3. Discussion

The specificity of a given T-cell clone is defined by its unique TCR, generated by somatic recombination, and remains static once successful rearrangement has occurred because TCRs are not subjected to somatic hypermutation following antigen exposure [11]. The detection of monoclonal TCR gene rearrangements is indirect evidence of clonality and is an important adjunct in the diagnosis of TCL as TCR rearrangement patterns are conserved within TCL [2]. The majority of diagnostic laboratories adopt PCR-based assays in the assessment of clonality because they do not require much input DNA and can be performed using FFPE tissues. In recent years, many laboratories use the BIOMED primers for PCR-based clonality assays as they are commercially available, standardized, and carefully designed to capture the rearrangement patterns of the vast majority of lymphomas bearing productive TCRs [3]. However, PCR-based clonality techniques are subjected to amplicon size restrictions, and the large size of the TCRA locus limits its assessment in the clinical setting. The TCRB gene, although smaller in size compared to TCRA, has a large germline-encoded repertoire and the resultant combinatorial diversity of TCRB gene rearrangements is extensive [3]. Therefore, only a subset of the TCRB locus is covered by the BIOMED multiplexing approach. For these reasons, the BIOMED-based T cell clonality assay is insufficient to fully characterize a TCR clonotype. Interpretation of the PCR-based assay is also dependent on the demonstration of a dominant amplicon of specific molecular weight and this strategy may be susceptible to interpretative error [12,13,14,15,16].

To our knowledge, the use of CNV analysis and the recognition of monoclonal patterns of TCR losses in the determination of T-cell clonality has not been studied. Our unique segmentation algorithm and CNV computational approach specifically identified 98% of TCLs with one or more TCR gene MRs using WGS data, slightly superior to the 91% PCR detection rate. With regards to sensitivity, this platform is able to detect MRs of TCR in the majority of samples with tumor content less than 25%, including cases with tumor content as low as 13%. This novel approach not only permits the computation of absolute CN data, thereby allowing the distinction of monoallelic from biallelic loss; it also enables a complete interrogation of all four TCR genes, unlike traditional PCR-based approaches which do not assess the TCRA gene on account of its complexity and amplicon size restrictions [3,5]. Moreover, we have demonstrated proof of concept that this strategy is potentially and readily applicable to the analysis of B-cell receptor loci for the determination of B-cell clonality.

During T-cell differentiation, the TCRD genes rearrange first, followed by TCRG, TCRB, and TCRA in a sequential manner. However, in reality, we do not necessarily always detect a linear sequence of rearrangement from TCRD to TCRA, and combinations of TCR gene rearrangement which do not follow this sequence can be detected by PCR [2]. This may be attributable to insufficient primer coverage resulting in a false negative result. Surprisingly, we also detected cases which did not demonstrate the expected sequential pattern of TCR rearrangement using our WGS CNV-based method, such as the presence of TCRB MRs in the absence of TCRA rearrangement, and vice versa. Validation with conventional PCR-based assay is not possible as the clonality assessment of TCRA loci is not routinely performed in diagnostic laboratories. Further study and correlation with NGS will be necessary to determine if our findings represent a true biological observation or technical limitation of our current platform.

Apart from T-cell lymphomas, clonal T-cell proliferations have been documented in elderly individuals, in patients with prolonged immune activation associated with chronic infection, tumors, autoimmune diseases, chronic alcoholism, hemochromatosis, and after allogeneic transplantation. Clonal T-cell expansions are also known to occur in myelodysplastic syndrome and conditions associated with the narrowing of T-cell repertoires, such as the acute phase of infectious mononucleosis and age-related thymus involution [17,18,19,20,21,22] While this knowledge is important to avoid a pitfall in the diagnosis of T-cell malignancies, the identification and characterization of clonal T-cell proliferations and the applicability of novel T-cell clonality assays are likely to extend beyond the scope of lymphoid malignancies.

Drawbacks of our current study include the small sample size and the inability of our platform to distinguish between different clones within the same tumor. While the conventional PCR-based clonality assay is able to detect more than one clone in a tumor, the determination of the number of clonal products for a given TCR locus and the number of T cell clones is not always straightforward [23]. First, the presence of two different PCR products is typically taken to reflect biclonality but may also be easily attributed to the occurrence of biallelic rearrangements in a single clone. Second, due to the specific configuration of the TCRB loci, multiple rearrangements can occur on one allele resulting in the possibility that up to four PCR products could still be compatible with a single clone. Moreover, the presence of multiple clonal TCR products could reflect biclonality, oligoclonality, or even pseudoclonality. This may explain some of the discrepancies in the biallelic vs. monoallelic rearrangements of TCRG and TCRB between PCR and our CNV-based approach.

Presently, there are several commercially available platforms that use NGS of PCR amplicons that can detect clonal T-cell proliferations, which include clonoSEQ^®^ (Adaptive Biotechnologies, Seattle, WA, USA) and LymphoTrack^®^ clonality assays (Invivoscribe, San Diego, CA, USA) and other emerging techniques [5,24,25,26,27]. These assays have high sensitivity and are able to discriminate between tumor clones; hence, they are useful in the detection of minimal residual disease. However, these modalities are amplicon-based assays and are dependent on the efficiency of the multiplexed PCR reactions and the primer design may not cover all possible rearrangements. In comparison, our WGS-CNV method cannot differentiate specific clonotypes but provides full coverage of all TCR and BCR loci. Therefore, there is less coverage bias using WGS than targeted sequencing. As WGS is increasingly being adopted for clinical use, the available genomics data represent a valuable resource which can be harnessed for clinical applications and the use of WGS data for clonality analysis is an additional feature that can be built into future clinical WGS pipelines. With the advent of NGS technology, the amount of input material and the cost of sequencing will be markedly reduced in the near future. Thus, it may be more cost effective to detect clonality using the WGS-CNV approach and we can also derive much more information from the whole genome data than just targeted sequencing of the TCR region alone. Our copy number-based approach to determine lymphoid cell clonality is also potentially applicable to FFPE samples, which form the majority of archival samples. One example is the Affymetrix Oncoscan CNV Assay kit, which is specifically designed to study copy number variation (CNV) in FFPE samples [28]. The input DNA is about 80ng and the resolution of the Oncoscan CNV Assay kit is as low as 50 kb for selected cancer genes. It is conceivable that a more targeted FFPE-based kit for the analysis of TCR rearrangements with even higher resolution can be generated by the utilization of an extended probe set focusing only on the TCR and BCR gene region in the future.

## 4. Materials and Methods

### 4.1. Case Selection

The study cohort consisted of forty-four samples of patients diagnosed with a variety of TCLs and ten samples of extranodal NK/T-cell lymphoma, nasal type (ENKTL) with adequate tissue material, which were obtained from the National Cancer Centre, Singapore General Hospital (Appendix A). Blood (*n* = 37) or buccal swabs (*n* = 7) of the same patient were used as matched controls. In all cases, there was no extensive involvement of the peripheral blood by disease. The TCLs selected included angioimmunoblastic TCL (AITL, *n* = 20), anaplastic large cell lymphoma (ALCL, *n* = 6), monomorphic epitheliotropic intestinal TCL (MEITL, *n* = 5), peripheral TCL not otherwise stated (PTCL, NOS, *n* = 12), and hepatosplenic gamma-delta TCL (HSTL, *n* = 1). In addition, 10 ENKTL cell lines (9 of NK-origin and 1 of T-cell origin) and one benign T-cell sample isolated from peripheral blood were also subjected to WGS (Appendix A).

The histologic diagnosis of the lymphomas was based on the 2017 World Health Organization Classification of tumors of hematopoietic and lymphoid tissues [29]. The diagnosis of ENKTL is based on the positive expression of CD3 and/or CD2, cytotoxic markers (TIA1 and/or granzyme B), and in situ hybridization for Epstein–Barr virus encoded small RNA (EBER-ISH), according to WHO criteria [29]. T-cell lineage was assigned based on a) positive expression of TCRB and/or TCRG protein by immunohistochemistry (IHC) and/or b) detection of monoclonal TCRG rearrangement by PCR. NK-cell lineage was assigned based one or more of the following features: (i) CD56+/CD8−/CD4− phenotype by IHC, (ii) lack of TCRB and TCRG expression by IHC and/or germline TCRG by PCR [6]. In our previous report, we have demonstrated that most of the T/NK cell lymphomas with CD8−/CD56+ phenotype were of NK-cell lineage, consistent with CD56 being a marker of NK-origin [6].

The study was approved by the National Healthcare Group Domain Specific Review Board (2009/00212).

### 4.2. Preparation of Genomic DNA for WGS

Genomic DNA was extracted from fresh-frozen tissue using DNeasy Blood & Tissue Kit (QIAGEN, Hilden, Germany) according to the manufacturer’s protocol. For whole genome sequencing, 200 ng of genomic DNA was fragmented by sonication and processed with library preparation following instructions by the TruSeq Nano DNA LT Library Prep Kit (Illumina, San Diego, CA, USA).

### 4.3. WGS and Determination of Copy Number Aberrations

Whole genome sequencing (WGS) was performed on matched tumor–normal samples on the Hiseq X sequencer (Ilumina, San Diego, CA, USA). The library was subjected to high throughput sequencing with the pair-end of 150 bp using the Hiseq X sequencer (Illumina, San Diego, CA, USA). Sequencing reads were aligned to the human 1k genome reference (hs37d5 and hg38) with BWA-MEM (v0.7.12). Sambamba (v0.5.8) was used to mark out the PCR duplicates. Somatic copy number alterations were predicted by CNVkit and interpreted as log2 ratios. CNVkit, with default parameters, was used to call copy number changes on the WGS data. The log2 ratio and CN are determined as follows: log2 ratio < −1.1 (CN = 0), log2 ratio −1.1 to −0.25 (CN = 1), log2 ratio −0.25 to 0.2 (CN = 2), log2 ratio 0.2–0.7 (CN = 3), log2 ratio >0.7 (CN = 4) (Figure 6). Further details are provided on CNVkit’s official website [30]. Reads aligned to the TCRD/G/B/A regions were extracted with SamTools (v1.2) for final visual confirmation.

### 4.4. Segmentation of Coverage Plots

The genomic positional coverage information for the TCR and IG genes was extracted using SamTools’s depth subroutine, and the sequencing depth information was then segmented by the “copynumber” R package [31] in Bioconductor. The segmentation process starts with a preprocessing step using the winsorize function to reduce the effects or outliers. The preprocessed data are then segmented using the PCF algorithm that segments single samples, each segment being a region of the gene which shows a unique sequencing depth. The segmentation analysis was carried out in R version 3.6.2.

### 4.5. Copy Number Prediction and Analysis

The aligned sequencing data from matched tumor–normal samples were used as input by CNVkit to predict segmented CN events. The subroutine “batch” was used to predict segmented copy number events, in terms of log2 (tumor_coverage/normal_coverage) and absolute CN, from the aligned WGS data [32]. For the cell lines, peripheral blood sample Nz3951 was used as the normal sample. Next, the subroutine “call” was used to perform absolute copy number calling on the output of “batch”. An absolute copy number deviating from 2 will be considered as a copy-change event.

Monoallelic CN loss was defined as CN = 1 and biallelic loss as CN = 0. The losses within each gene were then assigned into three categories based on the predicted CN and pattern of loss on the segmented plots—(i) monoclonal (flat pattern loss); (ii) polyclonal (slopy pattern loss); and (iii) no loss (Figure 2b).

### 4.6. WGS of DLBCL Cell Lines

The whole genome sequencing data of 10 DLBCL cell lines were downloaded from the public Sequence Read Archive (SRA) repository. The accession codes of the analyzed DLBCL cell lines are detailed in Appendix A.

### 4.7. Estimation of Tumor Content

Somatic single-nucleotide variations (sSNV) were curated and the median variant allele frequency (VAF) of the sSNVs was used to estimate the tumor content of the samples. For instance, a diploid sample will yield a tumor content estimate of 2*(median_VAF). Only tumoral samples of at least 13% were used for this study. The tumor content was categorized as follows: ≤20% (low), 21–40% (moderate), and >40% (adequate).

### 4.8. Sensitivity Testing Using In Silico Dilutional Analysis

To obtain the in silico sequencing data of SNT8, an NKTCL cell line of T-cell origin at different levels of dilution, WGS data were generated for SNT-8 (at 20%, 50%, and >90% tumor concentrations) and an NK-cell line. The two sets of data were then merged at different proportions to obtain in silico dilutions of SNT-8 at intervals of 10% concentrations using Samtools. The equation (Resultant = (a × A) + [(1 − a) × B]) governed how the data from the two chosen samples were split and remerged for the resultant data, where “A” represents SNT-8 and “B” represents the background (NK-cell line) sample, and “a” is the desired level of in silico dilution.

### 4.9. Analysis of TCRB and TCRG Clonality by PCR

Analysis of TCRB and TCRG clonality was performed on TCLs with sufficient DNA (*n* = 31) to benchmark the performance of WGS-based analysis. One-hundred twenty-five nanograms of DNA was isolated from prepared sections of formalin-fixed, paraffin-embedded tissue with the DNeasy Blood & Tissue Kit (Qiagen GmbH, Hilden, Germany), according to the manufacturer’s protocol. The PCR amplification was carried out using the TCRB + TCRG T-Cell Clonality Assay method using Biomed-2 primers for TCRG and TCRB (Invivoscribe Technologies’ IdentiClone™ assays, San Diego, CA, USA) according to the manufacturer’s protocols. PCR for TCR Beta gene rearrangement was performed using the following primer sets for Tube A (Vβ + Jβ1/2 primers), Tube B (Vβ + Jβ2), and Tube C (Dβ + Jβ1/2) and that for TCR Gamma using Tube A (Vg1–8, Vg10 + Jg 1.3/2.3) and Tube B (Vg9 and Vg11 + Jg 1.3/2.3). The products were run in a genetic analyzer using the ABI fluorescence detection method.

### 4.10. Targeted Sequencing of TCRB and Analysis

DNA-based targeted sequencing of the TCRB gene was performed using fresh-frozen tissue in selected cases (*n* = 12) with a read length of 2 × 150, in a subset of TCL cases using LymphoTrack TRB Assay Panel—MiSeq (Invivoscribe, San Diego, CA, USA) to verify TCRB rearrangement. A library was generated using 500 ng of genomic DNA with a one-step library preparation method according to the manufacturer’s protocol. The library was sequenced on a NextSeq 500 system (Illumina, San Diego, CA, USA) at the National University of Singapore with a read length of 2 × 150. Reads were aligned to the germline V, D, and J segments. Following alignment of the reads, complete and incomplete rearrangements were clustered using MIXCR [33] and Vidjil [34], respectively.

## 5. Conclusions

In summary, our study demonstrates the feasibility of utilizing a novel CNV-based approach to determine T-cell clonality in a sensitive and specific fashion, allowing the identification of MRs in TCR genes. Although extension to larger cohorts is warranted to confirm our findings and to test the platform on a wider spectrum of lymphoid proliferations for the assessment of false positivity, the proposed method represents a potentially valuable addition to current TCLs diagnostic assays and can also be applied to available WGS data of patients. Given that WGS is increasingly performed on FFPE specimens and optimized for routine clinical diagnostics [35], it is conceivable that the knowledge on the CNV data of B- and T-cell receptor genes can be further exploited in the design of novel markers of clonality in the foreseeable future.

## Figures and Tables

**Figure 1 cancers-13-00340-f001:**
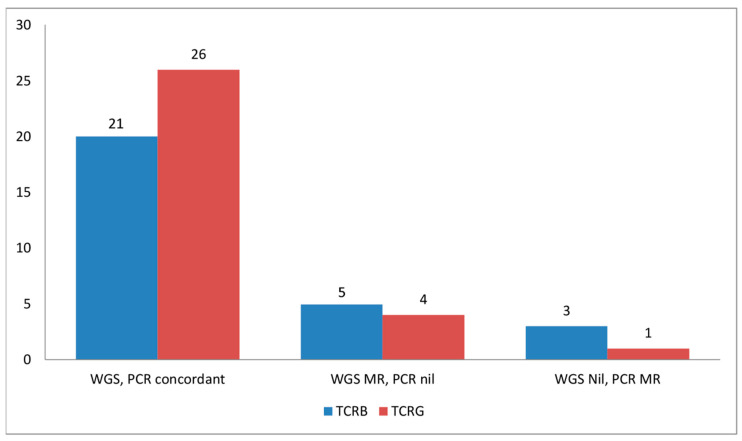
Comparison between detection of monoclonal rearrangements of the TCRB and TCRG gene as determined by whole genome sequencing (WGS) and PCR. PCR for TCRA is not available, while PCR for TCRD was not performed. Abbreviations: MR, monoclonal rearrangements; Nil, no monoclonal rearrangements.

**Figure 2 cancers-13-00340-f002:**
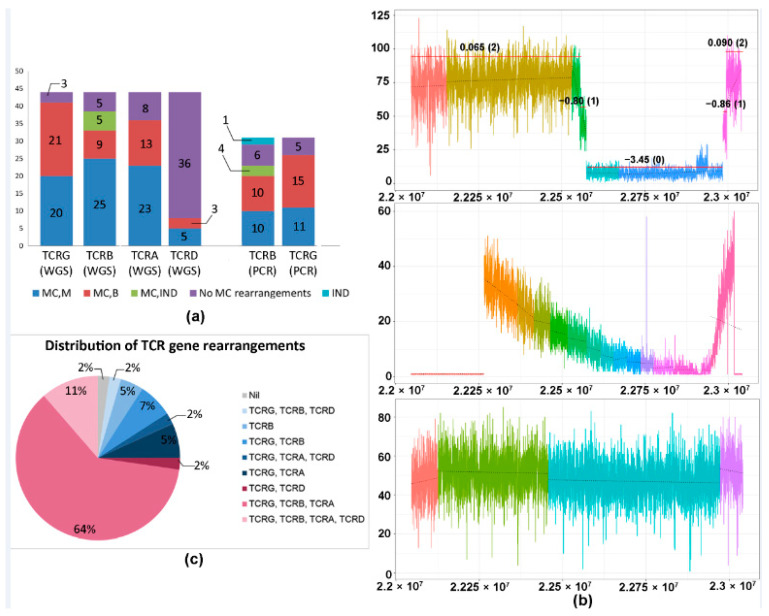
(**a**) Rearrangement status for each TCR gene as detected by WGS and PCR. Bar graph showing the nature (monoallelic/biallelic) and frequency of monoclonal rearrangements of each TCR gene, as determined by WGS and PCR. PCR for TCRA is not available, while PCR for TCRD was not performed. (**b**) Patterns of losses in the TCRA gene using WGS. (**Top panel**) Case of anaplastic large cell lymphoma (ALCL) showing a monoclonal pattern of TCRA loss with sharply demarcated broad segment deletions, indicating a clonal population with an identical copy number loss. Each region of the coverage plot flagged as a unique segment of the TCR gene by the segmentation algorithm is highlighted in a different color. Red horizontal lines represent regions of the coverage plots predicted to have a specific copy number variation by the copy number prediction algorithm. The degree of deviation from the aligned whole genome sequencing data, expressed in log2 ratios, is indicated above the red horizontal line for each segment and the corresponding absolute predicted copy number is indicated within brackets. In this example, the predicted copy number is “0”, indicating 2 copy (biallelic) loss. (**Middle panel**) Normal T-cells showing a polyclonal pattern in the TCRA gene with an overall “gradient”/“slopy” pattern of loss reflecting polyclonal T cells with diverse copy number losses across the gene. (**Bottom panel**) Extranodal NK/T-cell lymphoma (ENKTL) of NK origin showing no TCRA loss. (**c**) Pie chart showing the distribution of different patterns of rearrangements of TCR genes in cases subjected to WGS analysis. Abbreviations: MC, M, monoclonal, monoallelic; MC, B, monoclonal, biallelic; MC, IND, monoclonal, rearrangement indeterminate for monoallelic or biallelic; IND, indeterminate for monoclonal rearrangement.

**Figure 3 cancers-13-00340-f003:**
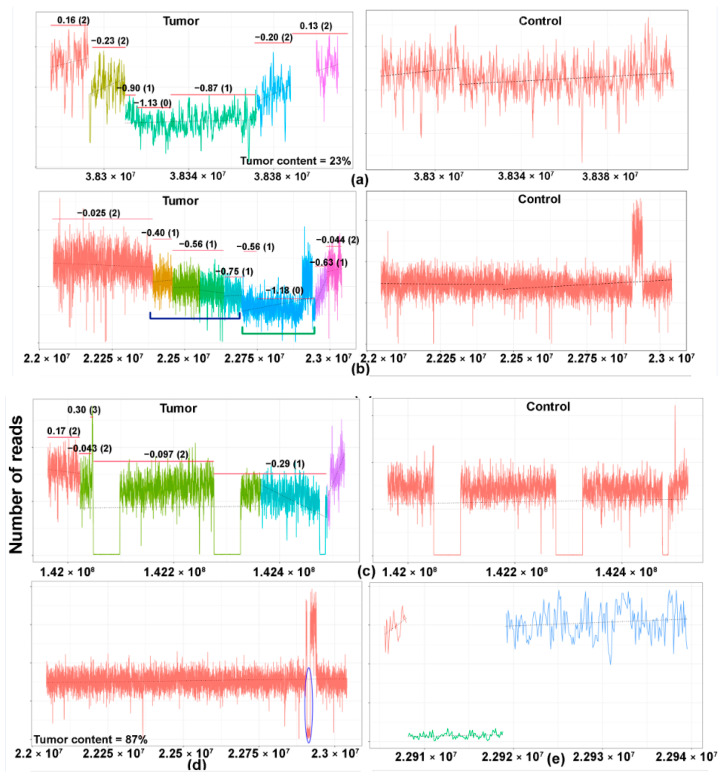
Copy number losses of TCR genes in T- and NK-cell lymphomas. The copy number profile of the TCR gene of the matched tumor (left) and controls (right) are illustrated in the coverage plots (**a**–**c**). The *Y*-axis represents the number of reads and the *X*-axis represents the location of various segments of the TCR gene. Each region of the coverage plot flagged as a unique segment of the TCR gene by the segmentation algorithm is highlighted in a different color. Red horizontal lines represent regions of the coverage plots predicted to have a specific copy number variation by the copy number prediction algorithm. The degree of deviation from the aligned whole genome sequencing data, expressed in log2 ratios, is indicated above the red horizontal line for each segment and the corresponding absolute predicted copy number is indicated within brackets. (**a**–**c**) A case of an angioimmunoblastic T-cell lymphoma (case 12) showing monoclonal rearrangements of (**a**) TCRG, (**b)** TCRA, and (**c**) TCRB. (**a**) Monoclonal pattern of loss of the TCRG gene with sharply demarcated broad segment deletions (green segment) compared to the matched control on the right, indicating a clonal population with a biallelic loss/rearrangement. (**b**) Monoclonal, biallelic rearrangement of the TCRA gene (blue segment, indicated by green bracket) occurring in a polyclonal background with a slopy gradient pattern (indicated in blue brackets). (**c**) Monoclonal, monoallelic loss of the TCRB gene (segment marked turquoise). (**d**,**e**) A case of angioimmunoblastic T-cell lymphoma (case 41) showing a very focal copy number loss (circled area) of TCRA (**d**), which represents actual monoclonal loss of the TCRD gene (**e**), green segment. (**f**) A case of angioimmunoblastic T-cell lymphoma (case 31) with no obvious monoclonal deletions. The blue circle demarcates the region magnified in (**g**). (**g**) Magnified coverage plot of the DJ region of the TCRB gene shows a broad segment deletion, corresponding to the incomplete TCRB rearrangement identified on sequencing.

**Figure 4 cancers-13-00340-f004:**
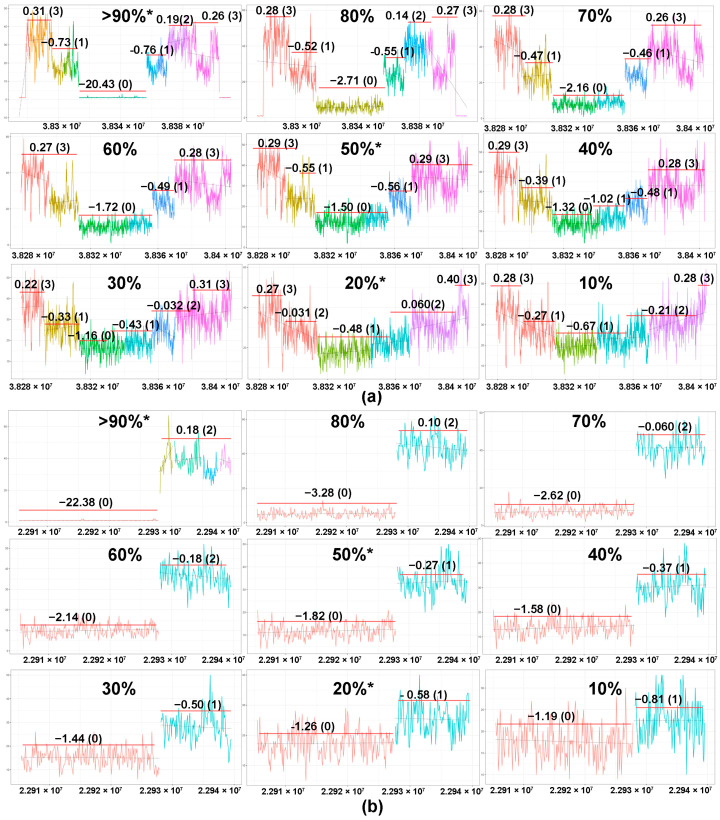
Sensitivity testing using in silico dilutional analysis. Detection of the monoclonal rearrangement of the TCRG and the TCRD genes of SNT-8, an NKTCL cell line of T-cell origin. WGS was performed on SNT-8 at 20%, 50%, and >90% tumor concentrations (marked with *). Utilizing a computational method, the predicted CN data for the other tumor concentrations were generated at 10% intervals. Based on the segmented plots and predicted copy number data, monoclonal rearrangements of TCRG ((**a**), green segment) and TCRD ((**b**), red segment) were detectable from >90% to 10% tumor concentrations. The magnitude of the drop in the log2 ratio of the coverage plot is correlated with the degree of dilution, indicating that the ability to predict the allelic status using WGS is affected by the sequencing depth at low tumor concentration.

**Figure 5 cancers-13-00340-f005:**
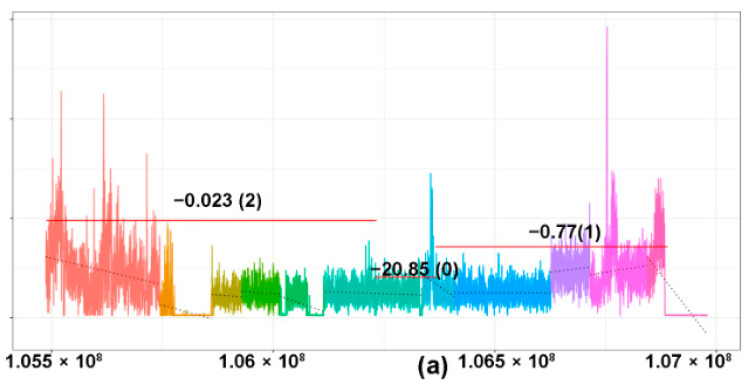
Copy number losses of BCR genes in the DLBCL cell line. (**a**–**c**) Examples of DLBCL cell lines showing monoclonal rearrangements of (**a**) IgH, (**b**) IgK, and (**c**) IgL.

**Figure 6 cancers-13-00340-f006:**
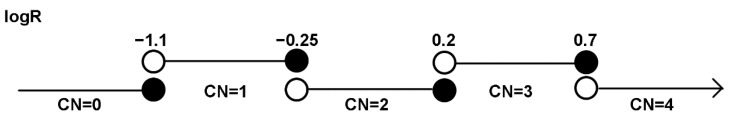
Cutoffs for log2 ratios used to determine copy number changes on WGS data.

**Table 1 cancers-13-00340-t001:** Top: Frequency and nature of monoclonal rearrangements of combinations and individual TCR genes in the WGS TCL cohort. Bottom: Frequency and nature of monoclonal rearrangements of TCR Genes in the PCR TCL cohort.

Frequency and Nature of Monoclonal Rearrangements of Combinations and Individual TCR Genes in WGS TCL Cohort				
**Combinations of Monoclonal TCR Rearrangements**	**Number of Cases with Particular Rearrangement (%) (*n* = 44)**			
Nil	1 (2)			
TCRB	2 (5)			
TCRG, TCRA	2 (5)			
TCRG, TCRB	3 (7)			
TCRG, TCRD	1 (2)			
TCRG, TCRB, TCRA	28 (64)			
TCRG, TCRB, TCRD	1 (2)			
TCRG, TCRA, TCRD	1 (2)			
TCRG, TCRB, TCRA, TCRD	5 (12)			
Total monoclonal cases	43 (98)			
**Frequency of Monoclonal Rearrangements of TCR Gene**	**TCRG (*n* = 44)**	**TCRB (*n* = 44)**	**TCRA (*n* = 44)**	**TCRD (*n* = 44)**
No. with monoclonal rearrangements (%)	41 (93)	39 (89)	36 (82)	8 (18)
No. without monoclonal rearrangements (%)	3 (7)	5 (11)	8 (18)	36 (82)
**Allelic Status of Monoclonal TCR Genes**	**TCRG (*n* = 41)**	**TCRB (*n* = 34) #**	**TCRA (*n* = 36)**	**TCRD (*n* = 8)**
No. of monoallelic rearrangements (%)	20 (49)	25 (73)	23 (64)	5 (63)
No. of biallelic rearrangements (%)	21 (51)	9 (27)	13 (36)	3 (38)
**Frequency and Nature of Monoclonal Rearrangements of TCR Genes in PCR TCL Cohort**				
**Combinations of Monoclonal TCR Rearrangements**	**Number of Cases with Particular Rearrangement (%) (*n* = 31)**			
Nil	3 (10)			
TCRB	2 (6)			
TCRG	3 (10)			
TCRG, TCRB	22 (71)			
TCRG, indeterminate for TCRB	1 (3)			
Total monoclonal cases	28 (91)			
**Frequency of Monoclonal Rearrangements of TCR Gene**	**TCRG (*n* = 31)**		**TCRB (*n* = 31)**	
No. with monoclonal rearrangements (%)	26 (84)		24 (77)	
No. without monoclonal rearrangements (%)	5 (16)		6 (19)	
No. with indeterminate rearrangements	0		1 (3)	
**Allelic Status of Monoclonal TCR Genes**	**TCRG (*n* = 26)**		**TCRB (*n* = 20) ##**	
No. of monoallelic rearrangements (%)	11 (42)		10 (50)	
No. of biallelic rearrangements (%)	15 (58)		10 (50)	

Abbreviations: TCR, T-cell receptor; TCL, T-cell lymphoma; TCRB, T-cell receptor beta; TCRG, T-cell receptor gamma; TCRA, T-cell receptor alpha; TCRD, T-cell receptor delta; #, excluding cases with only monoclonal rearrangements of DJ segment, where allelic status cannot be determined; PCR, polymerase chain reaction; ##, excluding cases with definite monoclonal rearrangements but indeterminate for monoallelic vs. biallelic rearrangements.

**Table 2 cancers-13-00340-t002:** Detection of monoclonal rearrangements of TCRG/TCRB via WGS and PCR.

Case No.	Case ID	TCRG MR on WGS	TCRG MR on PCR	TCRB MR on WGS	TCRB MR on PCR	Color Legend			
5	z3917	No	Yes	Yes	Yes	WGS Monoclonal, PCR No Rearrangement for TCRG			
3	z9951	Yes	Yes	Yes	No	WGS no rearrangement, PCR monoclonal for TCRG			
6	AITL13	Yes	Yes	Yes	Yes	WGS monoclonal, PCR no rearrangement for TCRB			
12	TR4323	Yes	Yes	Yes	Yes	WGS no rearrangement, PCR monoclonal for TCRB			
15	229A	Yes	Yes	Yes	Yes				
16	PTCL02	Yes	Yes	No	Yes				
18	T1443KC	Yes	Yes	Yes	Yes				
20	496A	Yes	Yes	Yes	Yes				
22	TR4241	Yes	Yes	Yes	Yes				
23	69A	Yes	Yes	Yes	No				
24	211A	Yes	Yes	Yes	Yes				
25	z4739	Yes	Yes	Yes	Yes				
26	z5286	Yes	Yes	No	Yes				
28	z3706	Yes	Yes	Yes	Yes				
29	TR4290	Yes	Yes	Yes	Yes				
30	AITL10	Yes	Yes	Yes	Yes				
32	SLS0697	Yes	Yes	No	IND				
33	z2756	Yes	Yes	Yes	Yes				
34	z8043	Yes	Yes	Yes	Yes				
36	z4110	Yes	Yes	Yes	Yes				
37	116A	Yes	Yes	Yes	Yes				
38	TR4232	Yes	Yes	Yes	No				
39	z3951	Yes	Yes	Yes	Yes				
42	z5469	Yes	Yes	Yes	Yes				
43	z5340	Yes	Yes	Yes	Yes				
31	194A	Yes	Yes	Yes	Yes				
8	TR4254	No	No	Yes	No				
9	z8343	Yes	No	Yes	No				
10	z5021	Yes	No	Yes	Yes				
17	z7285	Yes	No	Yes	No				
40	PB12311	Yes	No	Yes	Yes				
14	PB30818	No	NA	No	NA				
1	PB000933	Yes	NA	Yes	NA				
2	426A	Yes	NA	Yes	NA				
4	z6425	Yes	NA	No	NA				
7	193A	Yes	NA	Yes	NA				
11	z7889	Yes	NA	Yes	NA				
13	TR4313	Yes	NA	Yes	NA				
19	T1482TA	Yes	NA	Yes	NA				
21	TR4250	Yes	NA	Yes	NA				
27	62A	Yes	NA	Yes	NA				
35	TR4311	Yes	NA	Yes	NA				
41	226A	Yes	NA	Yes	NA				
44	T1473WK	Yes	NA	Yes	NA				

Abbreviations: TCRG, T-cell receptor gamma; MR, monoclonal rearrangements; WGS, whole genome sequencing; PCR, polymerase chain reaction; TCRB, T-cell receptor beta; IND, indeterminate; NA, Not available.

## Data Availability

The aligned sequencing data from the TCR genomic loci of the samples (Appendix A) can be assessed at Sequence Read Archive (SRA) under the BioProject ID: PRJNA692659.

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
