# Peer review of "T-Cell Lymphoma Clonality by Copy Number Variation Analysis of T-Cell Receptor Genes"

_cancers, 2021, doi:10.3390/cancers13020340_

Round 1 (The peer review reports for earlier submission)

Reviewer 1 Report

The manuscript presented by Oon and colleagues provides an evaluation of an analysis to determine the T-cell clonality of T-cell lymphomas using whole-genome sequencing (WGS). T-cell receptor rearrangement assays improve clinical diagnostics by demonstrating the presence of dominant clones. By bulk WGS, the authors showed monoclonal rearrangement of the T-cell receptor genes.

1.) Validation of mutations frequently observed in T-cell lymphoma by WGS would significantly strengthen this paper.
2.) Can the authors please clarify why they use WGS and not a targeted next-generation sequencing approach that gives a better sequencing depth and would be more achievable for clinical diagnostics
3.) What are the molecular signatures of the individual samples?
4.) Are there any differences observed in copy number variation between the different T-cell lymphomas (AITL vs. ILCL etc.) that could be used to diagnose lymphoid neoplasms?
5.) In Figure 2C, why does the control sample also show a monoallelic loss in TCRB?
6.) A major limitation is that the platform does not distinguish between different clones within the same tumor and this information has tremendous clinical value. Especially when 10x Genomics has kits available that allow you to look at VDJ rearrangement and gene expression on a single-cell basis.

Reviewer 2 Report

I really enjoyed reading this article and I commend the authors for their work in establishing an alternative route to establish T-cell clonality in the diagnosis of T-cell lymphomas.  The scientific methodology is sound overall - I do note that the PCR assay was from FFPE and it appears there was insufficient DNA in a minority of cases while the WGS analysis was from fresh-frozen tissue and the DNA threshold requirements appeared to be different too.

Recommendations:

  - In the introduction, I would like just a little bit more on the TCR biology -- maybe mention the basic structure of TCR (i.e. heterodimer of alpha/beta vs. gamma/delta) and that the diversity (D) segments are only found in the TCB/TCD loci.  Also I'd mention clonality is established fairly early in T-cell development since somatic hypermutation does not occur post antigen exposure -- while this is mentioned in the discussion, it may help to frame the context if at least briefly mentioned in the introduction.

  - I would remove the last line of the introduction -- this is more appropriate for the discussion.

  - As PB (vs. buccal swab) was used as the control, it may be worth mentioning that in the former cases (presumably) there was not extensive involvement of the PB by the lymphoma as some of these can present in a leukemia phase.  I would mention how many controls were PB vs. buccal swab.

  - DNA for the WGS was from fresh-frozen tissue and for the PCR was extracted from FFPE, and it looks like 200 ng DNA was obtained from the former and the threshold for the latter was 125 ng.  Only 31 cases were assessable by PCR.  This discrepancy in terms of the sample source / DNA threshold should be addressed more fully in the discussion (there is only a passing reference to increasing use of WGS with DNA extracted from FFPE).

  - As far as a future direction for this research, there are other instances where T-cell clonality is found and can be interested apart from T-cell lymphomas -- T-LGL, some MDS cases, other unexpected cytopenias, or a number of autoimmune diseases -- in short, other conditions where we know clonal T-cell proliferation can be found.

Reviewer 3 Report

This study proposes a method based on the analysis of copy number variation (CNV) from whole genome sequencing data to detect rearrangements of T-cell receptor (TCR) genes as a tool to assess clonality in T cell lymphoma. The authors showed that this strategy has higher detection rate compared to the PCR approach generally used in the clinic, and it can be used for samples in which the fraction of tumor cells is as low as 13%. The authors also make the point that this strategy is superior to PCR in that it enables to discriminate between monoallelic or biallelic deletions in all four TCR genes, including TCR alpha. While the approach is of potential interest, the comparison with other strategies is incomplete, making it difficult to fully appreciate its real advantages. Some criticisms are detailed below.

  1. Besides conventional PCR, other strategies, not mentioned by the authors, have been proposed to detect clonality in B and T cell malignancies, including NGS sequencing of PCR amplicons (e.g. Scheijen et al., Leukemia 2019) and targeted NGS sequencing (e.g. Kirsch et al., Sci.Transl.Med. 2015; de Masson et al., Sci.Transl.Med. 2018), for which a commercial kit is available. The authors need to explain the advantages of their method based on CNV analysis.
  2. How is the copy number estimation calculated? It would be useful to have some statistical analysis (for example, in Fig. 3b, for the 10% dilution the values -1.186 and -0.814 correspond to a copy number estimation of 0 and 1, respectively. What’s the degree of confidence?).
  3. Why in Fig. 3 the copy number is 3 for some regions?
  4. Table 1a and 1b should be combined to facilitate the comparison between their strategy and conventional PCR.
  5. The sequencing depth used for the analysis is not mentioned in the text. Is there a relationship between depth and the capacity to detect tumor cells at low concentration?
  6. The quality of the figures should be improved (e.g. resolution of Fig. 2g; text/values almost illegible in Fig. 1C and Fig. 3).
  7. The legend of Fig.2 is way too long. Part of it should be included in the results section.
  8. In the results section, paragraph 2.4 on B cell lymphoma cells is too short and it should be further developed. Also, the corresponding data should be included in a figure, not just shown as supplementary material.

Reviewer 4 Report

This paper describes a novel and useful approach to clonality detection using CNV detection.

It would be useful to have further details on the WGS coverage obtained and any suggestion whether smaller capture-based panels would be suitable for this approach/ what the requirements would be (e.g. would this work on FFPE samples?).

Fig 1 case 12: Why is there a 'slopy' polyclonal loss pattern observed in TRA but not in TRG if you state that this sample represents a clonal T cell population in a background of polyclonal T cells. What are the cut-offs  for CN1 and CN0?

You provide an example for a TRB D-J rearrangement that can be described to gene-level resolution based on the WGS data (Fig 2g)- is this feasible for all rearrangements types? This should be discussed and compared to the sequencing data obtained for TRB for those samples analysed with the Invivoscribe LymphoTrack NGS TRB assay. Please add details whether the DNA for the 12 samples run was derived from the FF or FFPE samples (section 4.10)

There is too little discussion with regards to those samples which showed discrepancies in the presence/absence of TR monoclonal rearrangements (And whether these were monoallelic or biallelic). I found the direct comparison provided in supplementary table 1 more informative than table 1a/1b in the main text and would suggest modification of table 1 to have a direct WGS-CNV versus PCR comparison.

E.g. for case 9 and 17 (PTCL/AITL, respectively) no evidence of clonality was found by PCR, whereas both cases are clearly clonal based on several targets by WGS-CNV. How could this be explained? What is meant by 'indeterminate' for the PCR assay?

E.g. for cases 16, 26 and 37 the TRB clonal rearrangements detected by PCR and amplicon-NGS are not picked-up by your WGS-CNV approach. How could this be explained?

For Fig 2d and 2f the circle is not visible and in Fig 2g the gene segments are very blurred. The tumour content should be provided for case 41 in Fig. 2

You detected no IGH/IGK deletions- was this concordant with PCR_based results given that 25-30% of AITL show clonal IG rearrangements?

Fig 3: "The magnitude of the drop in the log2 ratio of the coverage plot is correlated with the degree of dilution, suggesting that the ability to predict the allelic status using  WGS is partially dependent on tumor content." Please expand on this further to discuss the method's ability of detecting mono- and biallelic loss in samples with different neoplastic cell content. For the cell line experiments were the TRG and TRD mono- or biallelic losses as the figures show (0) and (1) depending on tumour infiltration level- this is not clear to the reader.

Supp table 3:case 37 and 16 are listed as CNV loss for TRB which contradicts supp table 2

Round 2 (The peer review reports for current submission)

Reviewer 1 Report

I now had a chance to review the updates the authors have provided in response to my comments and questions. I am satisfied with the clarifications and additions provided in the updated manuscript and would support its publication in the updated form.

Reviewer 2 Report

In their revised manuscript, the authors answered in a satisfactory manner to the different points raised during the previous round of review.

Reviewer 3 Report

I appreciate the edits made in the manuscript and vote to accept

Reviewer 4 Report

The authors have thoroughly revised the manuscript addressing most points adequately.